# Discriminative feature of cells characterizes cell populations of interest by a small subset of genes

**Takeru Fujii**[1,2☯], **Kazumitsu Maehara**[1☯]*, **Masatoshi Fujita**[2], **Yasuyuki Ohkawa**[1]*

**1** Division of Transcriptomics, Medical Institute of Bioregulation, Kyushu University, Fukuoka, Japan,
**2** Department of Cellular Biochemistry, Graduate School of Pharmaceutical Sciences, Kyushu University, Fukuoka, Japan

☯ These authors contributed equally to this work.
* kazumits@bioreg.kyushu-u.ac.jp (KM); yohkawa@bioreg.kyushu-u.ac.jp (YO)

**Data Availability Statement:** The codes used in our research are available at: https://github.com/tfwis/DFC.

**Funding:** This work was supported by Core research for evolutional science and technology

## Abstract

Organisms are composed of various cell types with specific states. To obtain a comprehensive understanding of the functions of organs and tissues, cell types have been classified and defined by identifying specific marker genes. Statistical tests are critical for identifying marker genes, which often involve evaluating differences in the mean expression levels of genes. Differentially expressed gene (DEG)-based analysis has been the most frequently used method of this kind. However, in association with increases in sample size such as in single-cell analysis, DEG-based analysis has faced difficulties associated with the inflation of P-values. Here, we propose the concept of discriminative feature of cells (DFC), an alternative to using DEG-based approaches. We implemented DFC using logistic regression with an adaptive LASSO penalty to perform binary classification for discriminating a population of interest and variable selection to obtain a small subset of defining genes. We demonstrated that DFC prioritized gene pairs with non-independent expression using artificial data and that DFC enabled characterization of the muscle satellite/progenitor cell population. The results revealed that DFC well captured cell-type-specific markers, specific gene expression patterns, and subcategories of this cell population. DFC may complement DEG-based methods for interpreting large data sets. DEG-based analysis uses lists of genes with differences in expression between groups, while DFC, which can be termed a discriminative approach, has potential applications in the task of cell characterization. Upon recent advances in the high-throughput analysis of single cells, methods of cell characterization such as scRNA-seq can be effectively subjected to the discriminative methods.

## Author summary

Statistical methods for detecting differences in individual gene expression are indispensable for understanding cell types. However, conventional statistical methods, such as differentially expressed gene (DEG)-based analysis, have faced difficulties associated with the inflation of P-values because of both the large sample size and selection bias introduced by

(JPMJCR16G1 to Y.O. https://www.jst.go.jp/kisoken/crest/en/index.html), Precursory Research for Embryonic Science and Technology (JPMJPR2026 to K.M. https://www.jst.go.jp/kisoken/presto/en/index.html) and Japan society for the promotion of science (JP18H04802, JP18H05527, JP19H05244, JP20H00456, JP20H04846, JP20K21398, and JP21H00232 to Y. O.; JP19H04970, JP19H03158, JP20H05393 and JP21H05755 to K.M. https://www.jsps.go.jp/english/). The funders had no role in study design, data collection and analysis, decision to publish, or preparation of the manuscript.

**Competing interests:** The authors have declared that no competing interests exist.

exploratory data analysis such as single-cell transcriptomics. Here, we propose the concept of discriminative feature of cells (DFC), an alternative to using DEG-based approaches. We implemented DFC using logistic regression with an adaptive LASSO penalty to perform binary classification for the discrimination of a population of interest and variable selection to obtain a small subset of defining genes. We demonstrated that DFC prioritized gene pairs with non-independent expression using artificial data, and that it enabled characterization of the muscle satellite/progenitor cell population. The results revealed that DFC well captured cell-type-specific markers, specific gene expression patterns, and subcategories of this cell population. DFC may complement differentially expressed gene-based methods for interpreting large data sets.

## Introduction

Organisms are composed of various cell types with specific functions. The cell types also include undifferentiated cells, such as stem cells and progenitor cells, or cells in transition during differentiation. Understanding cell types as the functional or structural units of an organism can confer a comprehensive understanding of the functions of organs and tissues, as well as their origins. The classification and definition of cell types have been based on the identification of specific marker genes that define them. Marker genes have been identified by comprehensive analysis of gene expression. Statistical tests are particularly important in the identification of marker genes, for which evaluation of differences in the mean expression levels of genes is often used.

Cell-type-specific genes/proteins are responsible for cell-type-specific functions. Therefore, to identify marker genes, comparisons have been performed between the cell types of interest and control groups to extract specifically expressed genes. The use of differentially expressed genes (DEGs) is a widely accepted way of defining marker gene candidates, the validity of which has been confirmed by biological experiments. However, the risk of false positives is increased by the tens of thousands of statistical tests associated with comprehensive analysis. Therefore, methods of correcting for multiple testing, such as Benjamini–Hochberg's false discovery rate (FDR) [1], Storey's q-value [2,3], and Efron's local FDR [4], have been widely employed. Along with the increasing demand for multiple testing and correction methods, DEG detection methods in the field of biostatistics for high-dimensional data have also been developed. In particular, limma [5], using Bayesian statistics, edgeR [6], and DESeq1-2 [7,8] have been developed to improve the statistical power (sensitivity) of DEGs while suppressing false positives. These methods have often been applied to cases in which only a small number of samples can be obtained because of the experimental scale and cost limitations [9].

Nevertheless, with the development of comprehensive gene expression analysis, especially single-cell analysis [10], new challenges have arisen as a result of the rapid increase in the number of samples [11] and the involvement of exploratory analysis schemes. The presence of a large sample size along with the application of exploratory analysis inducing selection bias can lead to an overly small P-value, impeding the application of conventional methods to call differentially expressed genes (DEGs) for bulk RNA-seq [12]. In particular, because a larger sample size can detect increasingly small differences, with a fixed difference, a larger sample size usually gives smaller P-values, resulting in the unnecessary expansion of candidate genes (e.g., the definition of the $t$-statistic is proportional to $\sqrt{n}$). Therefore, efforts to improve the ability to detect DEGs are still being made to adapt to scRNA-seq data with the characteristics of low coverage and large sample size. That is, one of the major problems to be solved in this field is

gene prioritization, namely, the selection of a small list of genes that should be validated and interpreted as a priority.

Here, we propose discriminative feature of cells (DFC), an alternative approach to the use of DEGs for characterizing cell populations by discrimination and variable selection. Ntranos et al. [13] incorporated logistic regression to detect isoform changes of transcript from scRNA-seq data. We further pursue the applicability of a discriminative approach for characterizing cell groups, especially for tissue scRNA-seq data in which gene expression levels are correlated and revealing the heterogeneous subpopulations within a specified population of interest (POI). We demonstrated that DFC succeeded in selecting a small set of genes that characterize a group of cells of interest, while avoiding the problem of a large candidate gene list due to the large sample size of scRNA-seq. DFC is also shown to have the potential to provide biological insights that are difficult to make using DEGs, such as detecting genes that characterize small subpopulations and specific combinations of genes that are functionally linked to each other.

## Results

We focused on the discriminative method to characterize a POI in cells (e.g., stem cell population), using scRNA-seq data from samples that contain a large number of cell types, such as tissues. In the discriminative method, each cell is considered as a data point in the gene expression space, and the decision boundary surface that separates the two groups is determined. The boundary surface is confined to a small dimension of the space by variable selection, which suggests the idea of characterizing cell populations by a selected subset of gene expression patterns as DFC (Fig 1A). While the conventional concept of using a DEG-based approach involves the comparison of group means of individual gene expression levels in two cell populations, the POI and a control group, DFC obtains a group of genes that are useful to discriminate the POI from the control group. Thus, DFC is supposed to provide a highly selective gene list of only the number of genes needed for discrimination, while simultaneously using information from all genes. In this study, we implemented the method for determining DFC using binary classification by logistic regression and variable selection by adaptive LASSO [14]. Specifically, we performed adaptive LASSO–logistic regression with the objective variable of belonging to the POI (1 or 0) and the explanatory variable of gene expression level; the genes whose weights were not 0 were considered as DFC.

First, we explored which characteristics of the POI are considered important by DFC. Because DFC could be related to multiple variables, it could potentially distinguish correlated gene pairs or mixed subpopulations. Therefore, we generated artificial scRNA-seq data assuming the above two scenarios of gene expression pattern that may be prioritized in DFC. We applied variable selection using adaptive LASSO–logistic regression on these artificial data to examine the characteristics of gene expression patterns of cell populations that tend to be DFC.

### DFC calls a pair of genes with dependences in their expression

In the first case, we tested the possibility that adaptive LASSO would prioritize correlated gene pairs. As the gene expression pattern of a simulated cell population, the expression levels of four genes were generated from a normal distribution with equal differences in group means and identical variances (Fig 1B). Therefore, all genes are equivalent as DEGs (i.e., they have the same P-value in the two-tailed t-test). However, only the gene pair $X_3$–$X_4$ is correlated ($r = 0.7$) within the two groups, while all other pairs are uncorrelated ($r = 0$). The expression of these hypothetical genes is shown in Fig 1C as a scatter plot. The process of variable selection (LASSO solution path) is shown in Fig 1D, indicating that, in the process of increasing the

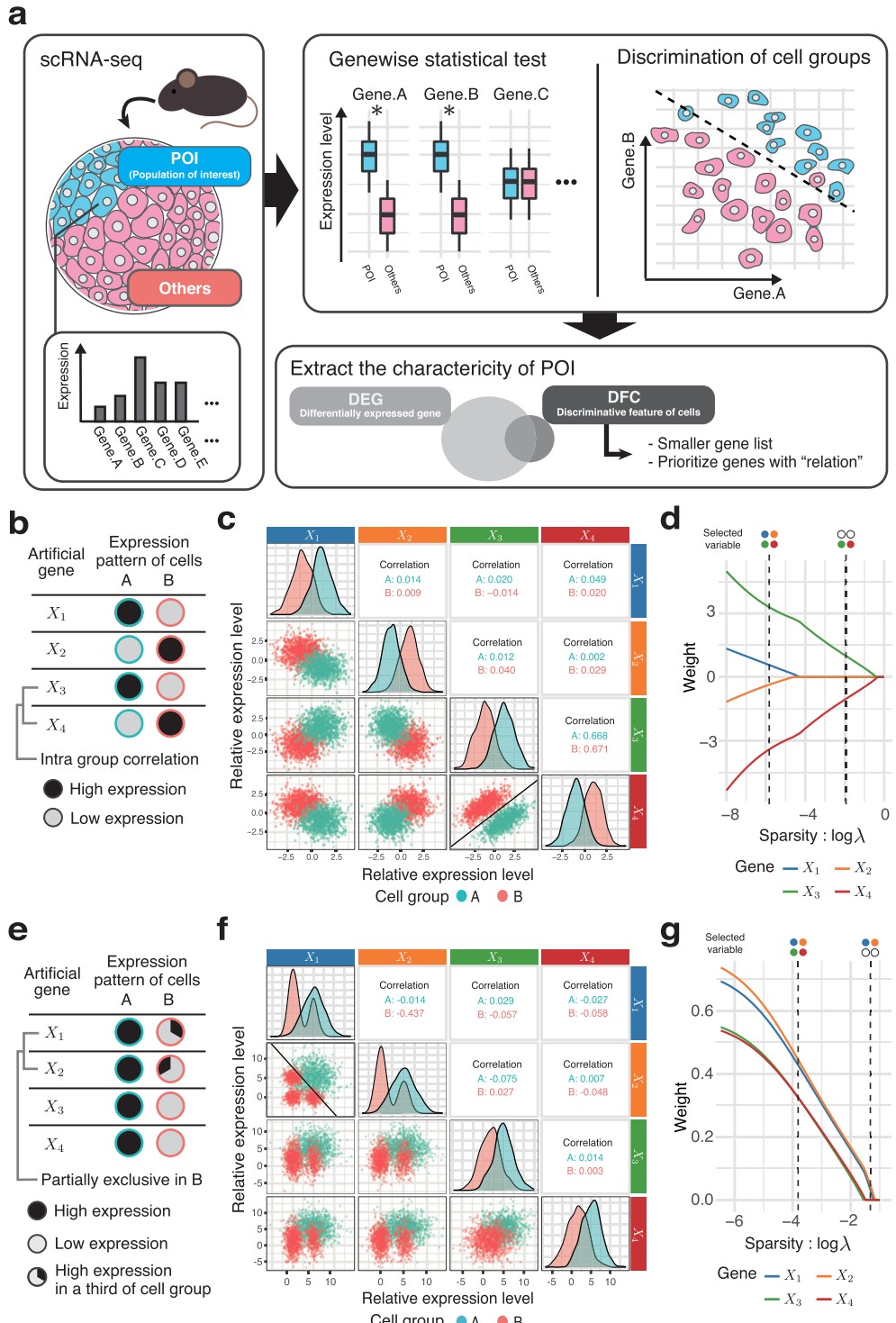

**Fig 1. The dependent pairs of gene expression selected as the DFC.** (a) Different concepts for gene selection of DFC and DEG. The common goal is to extract a set of genes that characterizes the population of interest (left). A DEG-based approach involves a list of genes with statistically significant differences between the studied groups. In contrast, the DFC-based approach involves a subset of genes that distinguish between two populations (top-right). DFC is expected to feature a small set of genes selected by taking into account the relationships among genes (bottom-right). (b–d) Artificially generated data set in which DFC has priority over DEG; case 1: correlation. (b) Schematic of the synthesized data design. Only the pair $X_3$ and $X_4$ has intra-group correlation; the other pairs are independent. All

variables have the same variance, and the differences in means are the same for all pairs (see Materials and Methods for details). (c) Pairs that are easier to classify are given priority to become DFC. The lower triangle shows the plot of each pair of variables; the diagonal elements show the distribution of each variable and the upper triangle shows the correlation coefficient within the cluster of each two variables. The decision boundary in the plain of the selected variable pair $X_3$ and $X_4$ is shown as a solid line. (d) The process of selecting discriminative variables; solution path. This indicates transition of the weights (partial regression coefficients) of each variable when regularization parameter $\lambda$ (sparsity) is varied. (e–g) Synthesized data set in which DFC has priority over DEG; case 2: exclusive. (e) Schematic of the synthesized data design. In one-third of the group A cells, the expression of $X_1$ and that of $X_2$ are mutually exclusive. The variances and the means of variables are designed as in case 1. In other words, this simulates a logical product relationship such that cells that express $X_1$ and $X_2$ simultaneously are equivalent to the population of group A. (f) An example of logical relationships of case 2, shown in a scatter plot as in (c). (g) The solution path in case 2 as shown in (d).

parameter $\lambda$, which adjusts the sparsity (the strength of variable selection), the correlated $X_3$–$X_4$ pair is finally selected (Fig 1D). This correlated pair has the clearest boundary between the groups (Fig 1C, bold box), and it can be intuited that the two selected variables are useful for differentiating groups A and B. To calculate the slope and intercept of the decision boundary in Fig 1C, we used the minimum value of $\lambda$ that gives two variables are selected. In addition, the accuracy of the discrimination using all four variables is 0.999, while the accuracy using only the two selected variables is 0.995, indicating that the model maintains adequate performance. These results indicate that gene pairs that are correlated within a group are likely to be selected as DFC. This suggests that DFC can select a set of genes that have direct or indirect dependences as useful features for discriminating groups.

We also investigated the nature of DFC using simulated data that more closely resemble real scRNA-seq data. We used ESCO [15], a simulator that takes drop-out into account, to generate the data. We designed the data with two cell populations and 15 genes. The two cell populations each have two marker genes. Two sets were generated and combined to create a data set containing gene sets derived from the two gene correlation networks (GCN) (500 cells for groupA and 493 cells for groupB) (S1A Fig). Gene1_1, Gene6_1, Gene7_1, and Gene8_1 were marker genes generated from GCN1, while Gene1_2, Gene4_2, Gene5_2, and Gene7_2 were ones generated from GCN2. The correlation between marker genes generated from different GCNs is low. Using these data, we extracted the DFC for Group1. In this case, we controlled the sparsity ($\lambda$) and limited the number of variables to be extracted to four. As a result, Gene1_1, Gene6_1, Gene1_2, and Gene4_2 were extracted. From the scatter plot, we can see that the pairs of Gene1_1 and Gene6_1, and Gene1_2 and Gene4_2 are highly correlated within the class and have high separability. Therefore, through the simulation by ESCO, we can show that the gene pairs with the same relationship as in Fig 1B, 1C and 1D are preferentially extracted.

Next, we examined the ability of DFC to detect mixed subpopulations. Here, we assume a situation that includes multiple subpopulations with different expression patterns in one group (Fig 1E). As in the previous scenario, all genes are equivalent as DEGs, that is, they have the same variance, and the group means are the same for all variables (see Materials and Methods). Here, genes $X_1$ and $X_2$ are strongly expressed in only one-third of the cells in group B. Furthermore, the expression of $X_1$ and that of $X_2$ are mutually exclusive. Therefore, $X_1$ and $X_2$ are not statistically independent. In addition, because group B is composed of multiple subpopulations, the distributions of gene expression of both $X_1$ and $X_2$ are multimodal (Fig 1F). The slope and intercept of the decision boundary were obtained in the same way as in Fig 1C. In this example, adaptive LASSO also prioritized the non-independent variable pair $X_1$ and $X_2$ (Fig 1G). This suggests that DFC is generally prone to selecting non-independent pairs of genes. In this scenario, the condition for being in cell group A is the expression of both $X_1$ and

$X_2$ at the same time, that is, the logical AND (&) relation, which cannot be realized by considering either $X_1$ or $X_2$ alone.

These results suggest that DFC may provide useful insights when considering cellular functions, not simply as a candidate list of differentially expressed genes, but as a set of genes that well defines characteristics of a cell population, including dependences such as correlations among multiple genes and mixtures of different populations.

## Small gene set of DFC determined by a unique criterion that differs from DEG-based analysis

Next, to demonstrate the practicability of DFC, we performed scRNA-seq data analysis and compared the yielded gene lists of DEG and DFC. The data were obtained from scRNA-seq data of mouse tibialis anterior (TA) muscle tissue injured by notexin at days 0, 2, 5, and 7 during the regeneration of skeletal muscle [16]. Skeletal muscle satellite cells (MuSCs) are known to be an essential cell population for muscle regeneration, and are activated upon muscle injury and undergo multiple progenitor cell stages until differentiating into myofibers. In this process of muscle regeneration, there is also a self-renewing state in which MuSCs again transition to a quiescent state and refill the MuSC pool [17,18]. Therefore, we attempted to characterize heterogeneous cell populations of MuSCs with multiple transient states by DFC, which have been difficult to capture by a DEG-based approach.

Fig 2A shows the procedure for defining the POI and the determination of DFC. First, public data of scRNA-seq (GSE143437) [16] were obtained. Then, the genes expressed in very few cells (fewer than 10 cells) were filtered (see Materials and Methods for details). Next, UMAP was used to visualize the data in two dimensions. The 12th cluster obtained by Louvain clustering (Fig 2B) was designated as the POI for this analysis, and all other clusters were designated as the control group (Fig 2C). Cluster 12 was selected according to the expression levels of *Pax7*, a marker gene for MuSCs, and *Myod1*, a transcription factor that functions in progenitor cells and activated satellite cells. The specific expression of the genes indicated that cluster 12 represents the satellite/progenitor cell cluster (Fig 2D and 2E).

Next, to elucidate the differences in the criteria for selecting genes in DFC and DEG in the data, we compared their gene lists. For DEGs, the criterion of FDR < 0.01 in DESeq2 was used. As a result of applying this criterion, the number of DEGs was 7,495, which was nearly half of the total (42%) of 16,351 mouse genes. Among the DFC, most of the genes (96/108) overlapped with the DEGs, but there were also 12 DFC-specific genes (Fig 2F). Next, we examined whether genes in DFC could be obtained by adjusting the gene selection criteria such as the P-value and log2 fold change in DEGs. Fig 2G shows the position of genes selected as DFC among all DEGs by a volcano plot. The large sample size of scRNA-seq and the statistical test on the clusters, which were also determined using the same scRNA-seq data, resulted in overly small FDRs. In addition, the genes selected as DFC among the DEGs were scattered irregularly. The results suggested that genes in DFC were selected independently of the DEG criteria. Similarly, in the MA (log ratio vs. abundance) plot (Fig 2H), genes in DFC were found to be scattered among the DEGs, indicating that the DFC selection was also independent of the gene expression levels. These results indicate that DFC selects genes according to its own criteria and also selects genes that are more useful for discrimination of the POI among the DEG candidates.

S1 Table summarizes the averages of the computational time and the number of extracted genes for DFC and DEG extraction for each method. In addition to the Sampling + adaptive LASSO–logistic regression and Wald test (DESeq2) used in this study, the results of the same analysis using five other methods are also shown. For DFC extraction, we repeated the analyses

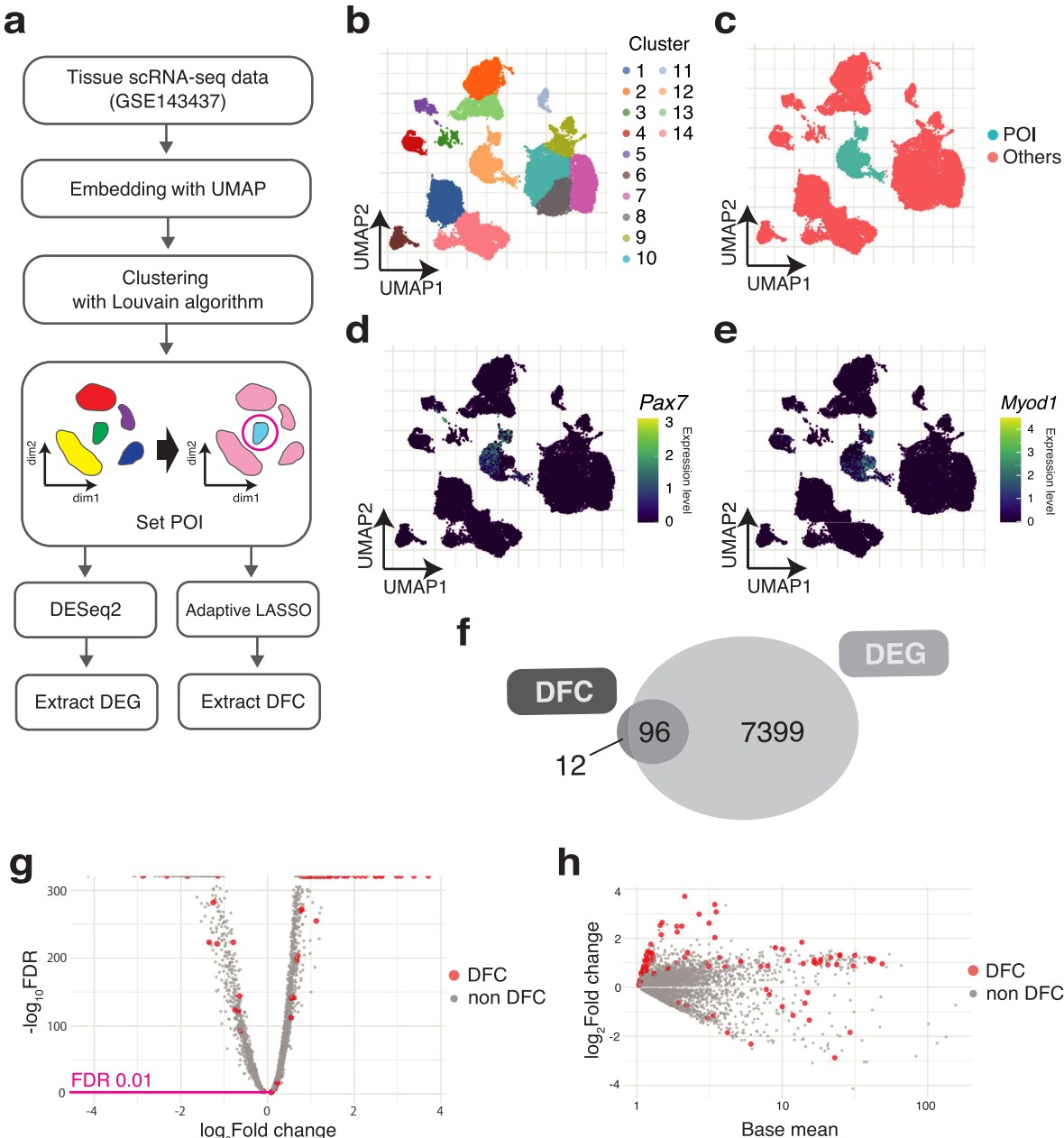

**Fig 2. Smaller gene set of DFC was selected by a unique selection criterion.** (a) Procedure of DEG and DFC extraction from scRNA-seq data. (b–e) The determined POI is compared with all other cell clusters in the muscle tissue. Embedding the scRNA-seq data into two-dimensional space with UMAP. (b) The clusters determined by the Louvain algorithm. The 12th cluster corresponds to the cluster of muscle stem cells and their progenitors. (c) The 12th cluster is set as the POI, and the other clusters are assigned as the control group, "Others." (d, e) Single-cell expression levels for *Pax7* and *Myod1*. (f) Some of the DEGs are selected as DFC. Venn diagram indicating the overlap of DEGs and DFC. (g) Genes in DFC not selected by the DEGs' criteria. Volcano plot of DEGs and (h) MA plot of DEGs.

ten times for averaging the effect of subsampling and cross-validation. For the other option to detect DEGs, limma [5] is known to be effective for DE analysis in single-cell RNA-seq data [10], the large number of genes was extracted as in DESeq2. For the DFC set extracted by each method, we created UpSet plots (S1C Fig) and analyzed the association between genes using the STRING database (S1 Table). The UpSet plots show that LASSO extracted a large number

of genes and contained many specific genes, and that there were many genes in common between Sampling + adaptive LASSO and SIS + adaptive LASSO. In addition, Sampling + adaptive LASSO showed the highest average number of edges (associations), while conversely this number was relatively low in SCAD. These results suggest that adaptive LASSO can extract a more biologically meaningful gene set in terms of the gene associations (e.g., co-expression, PPIs, pathways) than other discriminative methods.

In the extraction of DFC, the performance of SIS + adaptive LASSO was better than that of Sampling + adaptive LASSO in terms of computational time and the cross-validation error. Therefore, the use of SIS may be a promising option to reduce computational costs. However, in this study, we decided to use all genes by sampling cells to search a wide range of feature genes. The best hyper parameter $\gamma$ between 0.5 and 2.0 [14], which controls the size of the ridge penalty for adaptive LASSO, was determined using the faster SIS + adaptive LASSO. The number of selected genes decreased as the $\gamma$ increased, but the selected genes showed consistency (S1C Fig). In addition, the cross validation error was minimized at $\gamma = 1$. Therefore, we fixed at $\gamma = 1$.

The sample size ratio of POI to Other is 11.5, reflecting imbalanced data. This imbalance may cause a decrease in the performance of the discriminative model. We therefore evaluated the AUC of the precision–recall (PR) curve. The AUC was ~1 (S1D Fig), indicating that an appropriate model for discrimination had been created. We further examined the effect of increasing the imbalance between the sizes of POI and Other by training the discriminative model (extraction of DFC) using SIS + adaptive LASSO–logistic regression with a subsample of gradually decreasing POI size (S1D Fig). Even with a 10% subsample, the AUC was about 0.98, indicating that the model has sufficient ability to discriminate between groups. This can be attributed to the fact that we selected well-separable populations in this case of comparison after clustering. In line with the principles of logistic regression analysis for imbalanced data, it is worth checking the predictive performance using PR curves to be aware of any problem of overfitting. Such an overfitted model would overlook the characteristics of the POI.

## DFC is useful to identify the combinatorial patterns of gene expression and minor subpopulations

Next, we investigated whether DFC can extract the biological function of the POI. To evaluate this, we interpreted how genes in DFC help to discriminate the POI by referring to known marker genes of MuSC or muscle progenitors. First, we classified the DFC genes into three groups according to the specificity of expression in each cluster: The "Strong" feature refers to the genes expressed in 25% or more of the cells in one or two clusters. The "Weak" feature refers to the genes expressed in three or more clusters, as depicted in Fig 3A. The "Niche" feature is defined as genes with minor expression in less than 25% of the cells in all clusters (Fig 3B for the annotation of clusters, S2 Fig for the original annotation by the authors of the scRNA-seq data, and S4 and S5 Figs for highlighted expression of all genes in DFC as visualized using UMAP).

Genes belonging to the Strong feature included many of the genes known as markers of MuSCs and activated satellite cells (Figs 3C and S4A). M-cadherin (*Cdh15*) [19] was expressed almost universally in the POI. The results of GO enrichment analysis using only the Strong feature showed that many genes are related to skeletal muscle cells in muscle tissue (Fig 3D). In addition, *Myf5* is a representative feature of the group of cells that are not represented by *Myog* and *Myod1* in the POI (S4A Fig). Thus, it can be interpreted that *Myf5* plays a different role from other myogenic regulatory factors [20]. These results indicate that DFC can select genes that correspond to single biomarkers, similar to DEG.

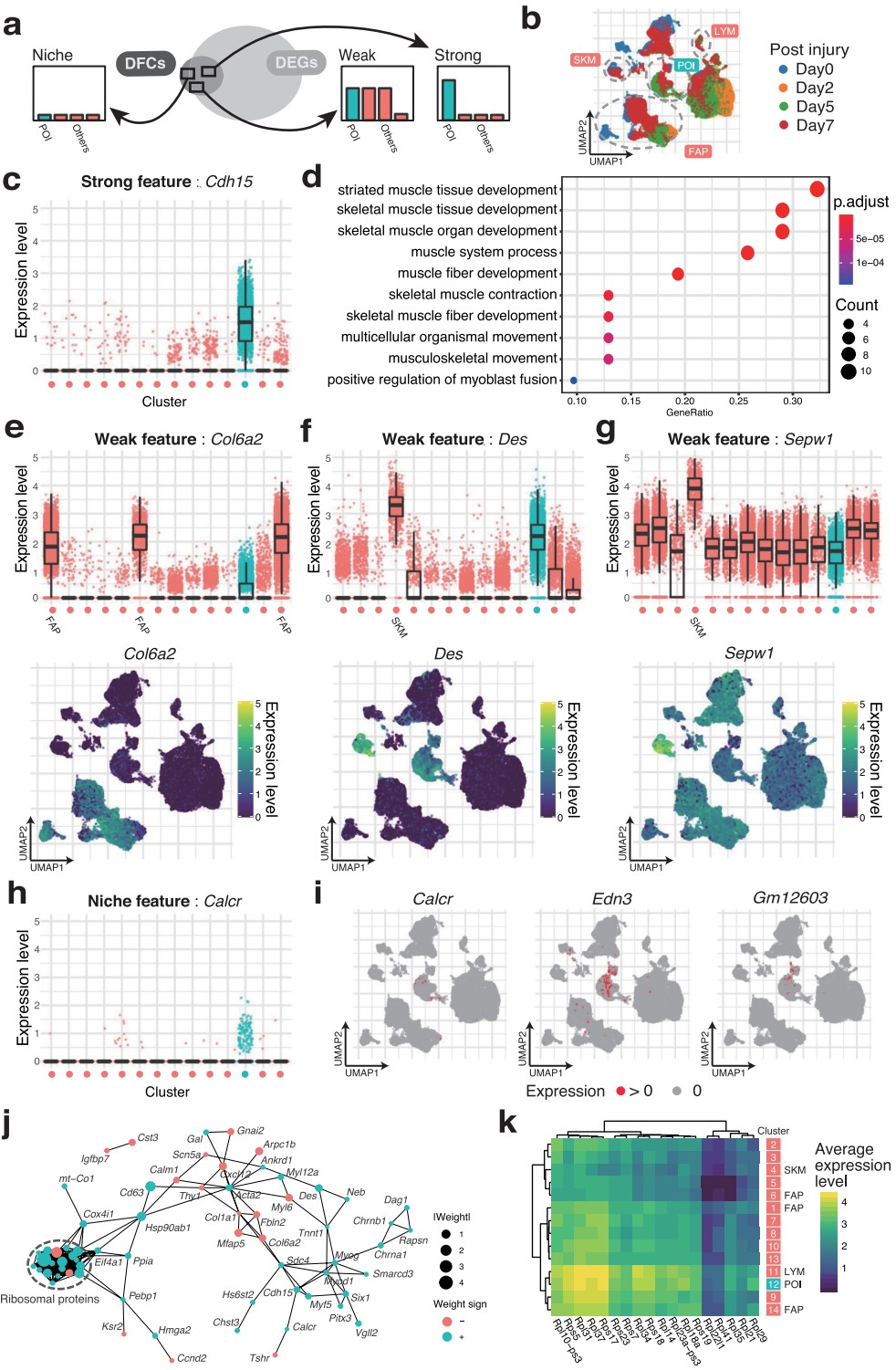

**Fig 3. Biological significance of genes in DFC revealed by the discriminative ability.** (a) According to the specificity of the expression, the genes in DFC are classified into three groups. The three groups are named Strong (specific to 1–2 clusters), Weak (>2), and Niche features (none of them). (b) The data are from samples collected at 0, 2, 5, and 7 days after skeletal muscle injury. In addition, the clusters of fibro/adipogenic progenitors (FAPs), mature skeletal muscle

(SKM), and lymphocytes (LYM) are shown. (c) Genes specifically expressed in the POI are assigned to the Strong features. For the Strong feature *Cdh15*, its expression level for each cluster shown in Fig 2B is plotted. The medians, 25th/75th percentiles, and 1.5 interquartile range (IQR) are employed to draw the box plots. (d) The Strong features contain many genes that act as markers of skeletal muscle. The results of GO enrichment analysis for the Strong features. GOs are ordered by the proportion of their inclusion in the Strong features. (e) Genes expressed in some clusters are assigned to the Weak features. For the Weak feature *Col6a2*, its expression level is plotted in the upper panel, and the single-cell expression level visualized by UMAP is plotted in the lower panel. (f, g) The expression levels of *Sepw1* and *Des*, two of the Weak features, are plotted as (e). (h) Genes with low expression levels that are expressed in a minor subpopulation of the POI are assigned to the Niche features. For the Niche feature *Calcr*, its expression level is plotted as (c). (i) Cells expressing Niche features (*Calcr*, *Edn3*, and *Gm12603*) are highlighted. (j) DFC has the property of capturing interrelated genes. STRING is used to connect related DFC. (k) Ribosomal proteins are a notablef example of Weak features in DFC that are difficult to interpret in the context of binary combinations. Eighteen ribosomal protein genes in DFC are averaged in each cluster as a heat map.

The Weak feature (S5 Fig) included Kai (*Cd82*) [21–23] and a group of genes used as quiescent satellite cell markers such as syndecan-4 (*Sdc4*). In contrast, the negative LASSO weight of *Col6a2*, which is selectively expressed in fibro/adipogenic progenitors (FAPs), helps to characterize the POI as non-FAPs (Fig 3E). Furthermore, a notable property of the Weak feature involved the combined expression pattern using multiple genes. For example, *Des* is expressed in both a part of the POI and the mature SKM population, which is a part of Others. This means that *Des* itself cannot characterize the POI (Figs 3F and S5). Therefore, to exclude the character of mature SKM from the POI, the condition of *Sepw1*(-), which is significantly expressed in mature SKM, is additionally imposed (Fig 3G). Thus, DFC can be used as a molecular marker to identify specific cell types for immunostaining and cell sorting, for example, *Des*(+)*Sepw1*(-)*Col6a2*(-).

The Niche feature detects minor subpopulations scattered within the POI (Figs 3H and S4B). These genes in DFC are difficult to prioritize in the list of DEG (i.e., they tend to have larger P-values). In fact, *Calcr*, which is ranked lower than 5,000th in the order of P-values in the DEGs, is known to be expressed transiently in a quiescent state [24,25]. We also detected *Edn3*[24], which has prominent localization on day 7 after injury in the POI, and *Gm12603* (*Wincr1*; WNT-induced noncoding RNA), a gene expressed in a different cell group from *Calcr* and *Edn3*.

These binary combinations (+/-) of multiple genes and minor cell groups would appear as the representative cases in Fig 1F. Therefore, DFC can characterize the best combinations of genes for determining cell type and even small subpopulations caused by transient expression or state changes, even using a small list of genes, and even upon comparison with the heterogeneous control group of cells in tissue.

To validate this discriminative approach of cell characterization with other data, we obtained the data of bone marrow, a tissue composed of relatively similar cells with the same lineage, as most of the cells in the bone marrow are differentiated from hematopoietic stem cells. Therefore, we thought that DFC could provide a more detailed evaluation. After visualization by UMAP and clustering by the Louvain algorithm, the hematopoietic precursor cell population was set as the POI (target) for analysis (S3A and S3B Fig). 31 genes were extracted, of which 14 were Strong features, 14 were Weak features, and 3 were niche features. Next, we attempted DE analysis by limma, and found 877 DEGs, of which two genes overlapped with DFC. Compared to the analysis of muscle tissue, the number of DEGs is small, but this may be due to the similarity of the cells that make up Bone marrow. Furthermore, these extracted genes were analyzed in detail by GO enrichment analysis. The GO enrichment analysis using strong features showed accumulation of myeloid cell differentiation-related terms such as interferon gamma production and T cell differentiation (S3C Fig). This result also suggests that POI is a population of hematopoietic progenitors related to lymphoid. In addition, three

genes were classified as Niche features, and *Socs1* was included as a characteristic gene (S3D Fig). *Socs1* has been reported to be a negative regulator of the JAK-STAT pathway [26], which is involved in the regulation of myeloid cell differentiation and proliferation [27,28]. It was thus suggested that *Socs1* is a factor that characterizes a small subpopulation with suppressed differentiation and proliferation.

## DFC extracts genes with functional associations

Finally, we attempted to elucidate more complex functional associations between genes from the obtained DFC. The artificial data in Fig 1 show that DFC selects the pair of genes with dependences among DEG-equivalent genes. This suggests that DFC may tend to contain the network of many-to-many gene relations that forms the unique characteristics of each cell type.

First, we examined the correlation matrix of expression levels for the genes in DFC, and found that the POI showed a more distinct hierarchical structure than the Others group (S6 Fig). We further evaluated the functional associations of 108 genes by STRING [29] (Figs 3J, S7 and S8). The 104 genes in DFC were contained in the STRING database and had 257 association edges, which is more than would occur by chance (87 edges), indicating that these are a set of genes that are strongly related to each other (PPI enrichment P-value $< 10^6$).

Next, we attempted to interpret the network by referring to the characteristics of the adaptive LASSO–logistic regression. First, we examined the correspondence with the weights of adaptive LASSO. We found that the two clusters of genes related to the function of FAPs with negative weights and the clusters of genes related to the activation of MuSCs with positive weights, such as *MyoD* and *Myog*, were linked via the proteoglycan *Sdc*4 (Fig 3J). Among them, the most remarkable result was obtained for the ribosomal protein-coding genes, which formed a dense cluster of ribosome subunit components. All of these genes are included in the Weak feature. The expression patterns of these genes are not clearly segregated into clusters and are difficult to interpret by binary combinations (S4 Fig), suggesting that they reflect a certain composition of ribosomal protein-coding genes that may have a critical functions in MuSCs and their progenitors. To confirm the discriminative ability of the genes, we extracted only the ribosomal protein-coding genes in the DFC, and visualized them by principal component analysis. The results confirmed that the subset of DFC was sufficient to discriminate the POI from the Others (S9A and S9B Fig). In more detail, the composition of these genes was characterized by higher overall expression levels compared with the Others, with *Rpl31* and *Rps37* being particularly highly expressed (Fig 3K). In contrast, the POI has a ribosomal protein profile closest to that of lymphocytes (LYM), but the negative weight of *Rpl34* appears to differentiate the POI from the lymphocyte population. Furthermore, these ribosomal protein-coding genes clearly captured the temporal changes after muscle injury in the POI (S9C and S9D Fig). This suggests that some ribosomal protein-coding genes are the critical factor in stem cell functions, such as self-renewal [30–32], and supports their role in muscle regeneration [33].

To further extend understanding of DFC for biological functions, we performed pathway analysis with the Reactome database [34,35] using all DFC (S2 Table). We found pathways enriched in ribosomal proteins such as rRNA processing and translation, and pathways enriched in skeletal muscle marker genes such as myogenesis. These results are consistent with the results of DFC set analysis by STRING (Fig 3J) and GO enrichment analysis using the Strong feature. Therefore, pathway analysis is also useful for investigating the nature of DFC or POI. We also examined how many DFC belong to esential genes, which are defined by Dickinson et al. as genes essential for survival. According to this definition, 5,280 genes were

essential genes [36], and 26.58% of DFC were essential genes in the data used in this study. In contrast, the number of essential genes in DFC was 32 ($32/108 \approx 0.30$). Therefore, there was no tendency for essential genes to be selected as DFC. Genes that are essential for survival do not need to be present as a characteristic of a particular POI. In conclusion, DFC can extract a small set of genes that characterize a POI, including functional associations between genes.

## Discussion

In this paper, we have proposed a new concept of characterizing the POI, which is an alternative to the DEG-based approach that uses lists of genes with differences in expression between groups. DFC has the potential to identify subpopulations within the population of interest and extract gene networks that regulate these subpopulations, in addition to the conventional identification of expression markers in cell populations, and is expected to significantly advance our understanding of cell populations from gene expression analysis. Our method, which can be termed a discriminative approach, has potential applications in the task of cell characterization. In particular, given the recent developments of high-throughput biological measurements, statistical models based on discrimination can be effective for the increased sample size of scRNA-seq (capable cell number).

To select a small number of genes to characterize a cell, as in a DEG-based approach, rather than to determine the cell type itself, a variable selection procedure was employed to select a small set of genes that are effective for discrimination. Variable selection is a methodology that selects a small number of $M < N$ optimal combinations of variables from $N$ input variables, while preserving the predictive performance of the statistical model. Several methods of variable selection with discriminative models have been developed, such as SVM [37], and logistic regression with LASSO penalty. Among them, LASSO–logistic regression is a method that can construct an interpretable linear model and perform variable selection in one step. However, the gene clusters obtained by these discriminative methods have been mostly used as gene signatures in cell-type classification [38,39] (e.g., a cell is normal or malignant), and no attempt has been made to interpret these gene signatures themselves biologically by giving rationales comes from the employed statistical method.

In this study, we compared two different population groups: the POI, which is specified after nonlinear dimensionality reduction and clustering, and the rest of the population. In this paper, this comparison was assumed to be the most frequently used procedure for profiling unknown cell populations using scRNA-seq data. However, DE analysis after clustering has been criticized for introducing selection bias, which results in excessively low P-values [12]. This exploratory data analysis of scRNA-seq makes the proper use of P-values more difficult, while our method bypasses the use of P-values. In addition, it can be assumed that a control group including heterogeneous populations will increase the variance in the group and lead to large P-values. For this reason, DE may miss subtle changes of state or fail to discover minor subpopulations within the cell groups of interest. In other words, calling DEGs may not be the best strategy for exploratory discovery in a mixed cell population represented by tissue. Furthermore, a simple two-group comparison of one vs. others is practical enough and thus is one of the major advantages of our method. The reason why this easy comparison works well is that DFC combines multiple Weak/Niche features to improve discriminative performance and pick up even small populations. The advantage is derived from the linearity of the model adapted in DFC; that is, the results can be interpreted as a superposition of features, as described above.

As an implementation of the concept of DFC, we employed the framework of binary classification with logistic regression and variable selection with adaptive LASSO. In addition to its

several beneficial statistical properties (e.g., consistency in variable selection), adaptive LASSO has superior practical performance among the improved versions of original LASSO [40]. Although there are many methods for variable selection, such as best subset selection (L0) [41], random forest [42], and SVM [37], in this study, we did not provide the benchmark tests of each method. However, we believe that how the mathematical properties of each method are used in the various scenarios of scRNA-seq data analysis is an important topic. In this paper, we have discussed the usefulness of the discriminative method when the dependence among all genes is included. We also found the discriminative approach to be useful, especially in the analysis of tissue scRNA-seq data where gene expression correlations and subpopulations within the same population are expected to be mixed. The further development of methods that focus on the interpretation of large-scale data is anticipated.

## Materials and methods

### Setting POI of scRNA-seq data

Normalized count matrix and the annotations of cells of scRNA-seq data were downloaded from GEO (GSE143437). We filtered out genes that were expressed ($> 0$) in fewer than 10 cells. UMAP visualization was performed using the *uwot* R package (version 0.1.10) [43]. In the embedded two-dimensional space, the clusters were determined by the Louvain algorithm [44] implemented in the *igraph* R package [45] (version 1.2.6). In our analysis of scRNA-seq data from muscle tissue, the POI was set as the cluster in which the majority of cells expressed both *Pax7* (53.3%) and *Myod1* (42.6%).

### Adaptive LASSO–logistic regression

The adaptive LASSO–logistic regression was performed using the *glmnet* [46] (version 4.1) R package. To perform the adaptive LASSO, we followed the two steps of parameter estimation: fitting the ridge and then fitting the LASSO regression with the penalty factor. These ridge (first step) and LASSO (second step) regressions in the adaptive LASSO were performed by setting the hyperparameter $\alpha$ in the glmnet::cv.glmnet function to 0 or 1, respectively. The sparsity parameter $\lambda$ was determined by 10-fold cross-validation (cv.glmnet) of binomial deviance. The penalty factor was set to be $1/|\boldsymbol{\beta}_{\mathrm{ridge}}|$, where $\boldsymbol{\beta}_{\mathrm{ridge}}$ is estimated by ridge regression in the first step. To reduce the computational cost in real scRNA-seq data, we obtained 30% subsamples of cells from each cluster (10,324 cells were extracted from 17,730 cells in total) in the estimation of $\boldsymbol{\beta}_{\mathrm{ridge}}$. In the performance comparison of SIS, LASSO and SCAD, *SIS* [47] and *ncvreg* [48] R packages were used (S1 Table).

### Differential expression analysis

A raw count matrix was downloaded from GEO (GSE143437). The Wald test was performed using the *DESeq2* [8] (version 1.28.1) R package to identify genes that were differentially expressed (DEGs) between the POI and the Others. The parameters were used with the default settings. Genes with FDR $< 1\%$ were considered significantly differentially expressed.

### Synthetic data generation

The artificial data set consists of randomly generated data points (cells) with four variables (genes). We set the variables as the *equivalent genes* in terms of DE. Because the statistical significance of a DEG that is estimated by the *z*- or *t*-statistic is uniquely determined by variances and the difference of means between groups, we only modified the relationships of the genes, while maintaining the variance and difference of means. Specifically, the two cases of DE-

equivalent genes were generated as follows. Case I: Correlated expression. All four variables follow the Gaussian distribution with constant variance $\sigma^2 = 1$ and difference of means $|\mu_A - \mu_B| = 2$, where A and B indicate groups of cells (each of 1000 cells). All pairs of variables are independent ($r = 0$), except for the pair ($X_3, X_4$) having a strong correlation ($r = 0.7$) in both groups. Case II: Heterogenous population. All four variables have the same variance $\sigma^2$ and the same group means $\mu_A, \mu_B$. We set $X_1$ and $X_2$ of group B as having an exclusive relationship. We divided group B into three subgroups (B1–3: each of 333 cells). Group B1 expresses $X_1$, group B2 expresses $X_2$, and group B3 expresses neither of them. The others are independent Gaussian variables. To equalize the variance and means in group B to those in the others, we used the mixed distribution as the marginal distribution of $X_1$ and $X_2$ in B:

$$f = pg(\mu_1, \sigma_1^2) + (1 - p)g(\mu_2, \sigma_2^2),$$

where $g$ is the Gaussian probability density function and $p$ is the proportion of subgroup relative to the size of group B. In general, the mean and variance of $f$ are calculated as follows:

$$\mu = p\mu_1 + (1 - p)\mu_2$$

$$\sigma^2 = p\sigma_1^2 + (1 - p)\sigma_2^2 + p(1 - p)(\mu_1 - \mu_2)^2.$$

We used the parameters: $\sigma_1^2 = \sigma_2^2 = 1$, $\mu_1 = 0$, $\mu_2 = 5$, $p = 2/3$, and hence $\mu_B = 5/3$ and $\sigma^2 = 59/9$ for all variables. We set $\mu_A = 5$.

## Computational environment

All calculations in this study were performed under the following environment.

- CPU: Intel Xeon Skylake-SP (2.3 GHz, 18 core) × 2

- Memory: 384 GB

## Supporting information

**S1 Fig.** (a) Pairs of marker genes that are simulated by ESCO. The lower triangle shows the plot of each pair of variables; the diagonal elements show the distribution of each variable and the upper triangle shows the correlation coefficient within the cluster of each two variables. The highlighted pairs are prioritized in DFC selection. (b) UpSet plot to compare DFC sets extracted by each four methods (SCAD, SIS + adaptive LASSO, Sampling + adaptive LASSO and LASSO). Each columns represents number of genes that are shared by only the marked sets. (c) UpSet plot to compare DFC sets extracted by SIS + adaptive LASSO with $\gamma = 0.5, 1, 2$. (d) As the ratio of A to B increases, the performance of the generated model deteriorates. The PR curve when the sample size of POI is gradually decreased (100%, 90%,. . ., 10%).
(PDF)

**S2 Fig. Original annotations of scRNA-seq data by the authors [GSE143437].** (a) Days after muscle injury. (b) Cell type annotations.
(PDF)

**S3 Fig.** (a) The scRNA-seq data embedded into two-dimensional space with UMAP. (b) "POI" and "Others" determined by the Louvain algorithm. POI corresponds to the cluster of hematopoietic precursor cells. (c) The Strong features contain many genes that are the markers of POI. The results of GO enrichment analysis for the Strong features. GOs are ordered by the contained proportion of Strong feature genes. (d) Zoom in on the area indicated in Fig S3b.

Cells expressing Niche features (*Calcr*, *Edn3*, and *Gm12603*) are highlighted.
(PDF)

**S4 Fig.** UMAP visualizations of (a) Strong and (b) Niche features. The markers colored in red/blue indicate the sign of the weight (coefficient) estimated by adaptive LASSO.
(PDF)

**S5 Fig. UMAP visualizations of Weak feature.** The markers colored in red/blue indicate the sign of the weight (coefficient) estimated by adaptive LASSO.
(PDF)

**S6 Fig. The correlation coefficient matrix of genes in DFC shows a distinct hierarchical structure within the POI.** The matrixes in (a) POIs and in (b) Others are shown. Types of feature (Strong, Weak, or Niche) and the signs of LASSO weight are also indicated.
(PDF)

**S7 Fig. Functional association of genes in DFC.** The functional associations among 108 DFC genes were annotated using STRING.
(PDF)

**S8 Fig. Functional association of DEGs.** The functional associations of the top 108 genes in terms of P-value among the DEGs annotated by STRING.
(PDF)

**S9 Fig. Some of the ribosomal protein-coding genes characterizing the POI.** Scatter plot showing the results of PCA performed using (a) ribosomal protein-coding genes in DFC and (b) all ribosomal protein-coding genes. (c) PCA performed in the POI using genes in (a), and in all cells (POI and others) using genes in (b).
(PDF)

**S1 Table. Summarizing the performance of DFC and DEG extraction method and the result of analyzing the association between DFC sets using the STRING database.**
(XLSX)

**S2 Table. Pathways enriched in DFC analyzed with Reactome database.**
(XLSX)

## Acknowledgments

Computations were carried out using the computer resources offered under the category of Intensively Promoted Projects by the Research Institute for Information Technology at Kyushu University. The authors thank Edanz (https://jp.edanz.com/ac) for editing the English text of a draft of this manuscript.

## Author Contributions

**Conceptualization:** Takeru Fujii, Kazumitsu Maehara, Masatoshi Fujita, Yasuyuki Ohkawa.

**Data curation:** Takeru Fujii.

**Formal analysis:** Takeru Fujii, Kazumitsu Maehara.

**Funding acquisition:** Kazumitsu Maehara, Yasuyuki Ohkawa.

**Investigation:** Takeru Fujii, Kazumitsu Maehara, Masatoshi Fujita.

**Methodology:** Takeru Fujii, Kazumitsu Maehara.

**Project administration:** Kazumitsu Maehara, Yasuyuki Ohkawa.

**Resources:** Kazumitsu Maehara, Yasuyuki Ohkawa.

**Software:** Takeru Fujii, Kazumitsu Maehara.

**Supervision:** Kazumitsu Maehara, Yasuyuki Ohkawa.

**Validation:** Kazumitsu Maehara.

**Visualization:** Takeru Fujii.

**Writing – original draft:** Takeru Fujii, Kazumitsu Maehara, Masatoshi Fujita, Yasuyuki Ohkawa.

**Writing – review & editing:** Takeru Fujii, Kazumitsu Maehara, Yasuyuki Ohkawa.

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
