## [Decision Letter · Decision Letter 0]

13 Jul 2021

Dear Prof. Ohkawa,

Thank you very much for submitting your manuscript "Discriminative feature of cells characterizes cell populations of interest by a small subset of genes" for consideration at PLOS Computational Biology.

As with all papers reviewed by the journal, your manuscript was reviewed by members of the editorial board and by several independent reviewers. In light of the reviews (below this email), we would like to invite the resubmission of a significantly-revised version that takes into account the reviewers' comments.

We cannot make any decision about publication until we have seen the revised manuscript and your response to the reviewers' comments. Your revised manuscript is also likely to be sent to reviewers for further evaluation.

Sincerely,

Lilia M. Iakoucheva, Ph.D.

Associate Editor

PLOS Computational Biology

Ilya Ioshikhes

Deputy Editor

PLOS Computational Biology

Reviewer's Responses to Questions

**Comments to the Authors:**

Reviewer #1: The authors have proposed using the Elastic Net package in R to address the limitation of differentially expressed gene identification. The idea looks legitimate. However, the technical contribution to methodology could be limited. There are several key questions which have to be addressed to fit into the preference of computational biology, on top of bioinformatics.

1. It is straightforward to expect that the adaptive LASSO can always identify a small set of key genes for cell type / population identification. Therefore, the DFC set looks smaller than the DEG. I was wondering what are the biological significance of that DFC set? Are those genes only used to differentiate cell populations? What are the implications behind the DFC set. Is the DFC set reflecting / overlapping with the "essential genes" ?

https://www.nature.com/articles/nrg.2017.75

2. The technical contribution to the methodology looks limited as it is only based on the off-the-book-shelf adaptive LASSO method from the well-known Elastic Net package in R. Comparisons to other feature selection or model regularization methods in machine learning could be beneficial.

3. The underlying pathways behind those DFC genes could be interesting.

4. The concept of DEG has been well-established in the past years. It is therefore important to tell the difference between DFC and DEG. The overlapping and non-overlapping genes between DFC and DEG should be analysed further to reveal its biological difference / similarity.

5. The running time and computing environment should be reported as gene expression data can be very big in scale.

Reviewer #2: In this manuscript, authors proposed a new concept of discriminative feature of cells (DFC), which was an alternative to the DEG-based approach that uses lists of genes with differences in expression. DFC was a discriminative approach implemented using adaptive LASSO logistic regression. The results revealed that DFC well captured cell-type-specific markers, specific gene expression patterns, and subcategories of the cell population. Some comments are given below.

1. For section “Adaptive LASSO–logistic regression” staring from line 380, the adaptive Lasso used was a little different from the method in reference [13]. For computations of adaptive LASSO, the gamma parameter in the weights or penalty factor was directly set to be 1. In the reference, the optimal pair of gamma and lambda parameters was obtained by cross-validation. It is suggested to explain why gamma was directly set to be 1.

2. Ridge regression estimates were used in the first step with 30% subsamples of cells from each cluster. Please give more details about the sample size and the number of variables involved. For ultrahigh dimensional statistical models, (I)SIS is suggested. And based on the code available on Github, Ridge regression estimates were chosen by cross-validation (cv.glmnet), which is not shown in the manuscript. It is suggested to give more details of the implemented adaptive LASSO.

Diego Franco Saldana and Yang Feng (2018) SIS: An R package for Sure Independence Screening in Ultrahigh Dimensional Statistical Models, Journal of Statistical Software, 83, 2, 1-25.

Jianqing Fan and Jinchi Lv (2008) Sure Independence Screening for Ultrahigh Dimensional Feature Space (with discussion). Journal of Royal Statistical Society B, 70, 849-911.

3. Since logistic regression was involved, it is suggested to consider the impact of imbalanced binary response data.

4. It is suggested to compare adaptive LASSO with other variable selection methods, e.g. SCAD.

5. It is suggested to demonstrate the computation efficiency of DFC methods. For example, the computation time comparison between DEG and DFC methods.

6. It is suggested to do another real data analysis to validate the results for DFC.

7. For the code on Github, string.nb.html is not displayed correctly.

Reviewer #3: This article develops a new method to identify cell-type-specific genes based on single-cell RNA sequencing (scRNA-seq) data. The basic idea is to fit a classification model between the population of interest and other groups, and the identification of marker genes is viewed as a variable selection problem. In this article, the authors promoted the use of a logistic regression model with adaptive Lasso penalty, and genes with nonzero regression coefficients are defined to be discriminative features of cells (DFC).

Overall, this article introduces a tool that could be useful in practice, but I would hope that the authors can address the issues I list below.

1. The authors should have provided a more comprehensive review of existing works. The use of discriminative models for differential expression analysis is not completely new. For example, [1] fits a logistic regression to predict cell membership, although their predictors are transcripts instead of genes. The authors need to explain the overlap with prior art and highlight the novelty.

2. The proposed method was mainly compared with DESeq2. However, DESeq2 was originally designed for bulk RNA-seq data, and scRNA-seq data have certain characteristics that may harm the performance of these methods. This phenomenon was explained in [2] in more details. Therefore, the authors may want to consider some methods that are tailored for scRNA-seq, for example the ones listed in Table 1 of [2].

3. The simulation of synthetic data can be made more realistic, as scRNA-seq data in real world contain more noise than the simple normal distribution used in the article. For example, the drop-out effect is one characteristic of scRNA-seq that should not be ignored. The authors may consider existing simulators such as Splatter [3] and ESCO [4].

4. Minor comment: the text "in principle the P-value decreases with increasing sample size" in line 79-80 is not very accurate. Under the null hypothesis (no difference), P-value follows a uniform distribution, so it does not decrease to zero as sample size increases. The actual meaning is that a larger sample size can detect smaller and smaller differences, so with a fixed difference, a larger sample size usually gives smaller P-values.

5. Minor comment: in Figure 1d and 1g, what are the actual lambda values? Does cross validation correctly select the sparse model?

[1] Ntranos, V., Yi, L., Melsted, P., & Pachter, L. (2019). A discriminative learning approach to differential expression analysis for single-cell RNA-seq. Nature methods, 16(2), 163-166.

[2] Wang, T., Li, B., Nelson, C. E., & Nabavi, S. (2019). Comparative analysis of differential gene expression analysis tools for single-cell RNA sequencing data. BMC bioinformatics, 20(1), 1-16.

[3] Zappia, L., Phipson, B., & Oshlack, A. (2017). Splatter: simulation of single-cell RNA sequencing data. Genome biology, 18(1), 1-15.

[4] Tian, J., Wang, J., & Roeder, K. (2020). ESCO: single cell expression simulation incorporating gene co-expression. bioRxiv.

**Have the authors made all data and (if applicable) computational code underlying the findings in their manuscript fully available?**

Reviewer #1: None

Reviewer #2: Yes

Reviewer #3: Yes

PLOS authors have the option to publish the peer review history of their article (what does this mean?). If published, this will include your full peer review and any attached files.

Reviewer #1: No

Reviewer #2: No

Reviewer #3: No
---

## [Decision Letter · Decision Letter 1]

9 Oct 2021

Dear Prof. Ohkawa,

Thank you very much for submitting your manuscript "Discriminative feature of cells characterizes cell populations of interest by a small subset of genes" for consideration at PLOS Computational Biology. As with all papers reviewed by the journal, your manuscript was reviewed by members of the editorial board and by several independent reviewers. The reviewers appreciated the attention to an important topic. Based on the reviews, we are likely to accept this manuscript for publication, providing that you modify the manuscript according to the review recommendations.

Sincerely,

Lilia M. Iakoucheva, Ph.D.

Associate Editor

PLOS Computational Biology

Ilya Ioshikhes

Deputy Editor

PLOS Computational Biology

[LINK]

Reviewer's Responses to Questions

**Comments to the Authors:**

Reviewer #1: The authors have addressed my comments.

Reviewer #2: This manuscript has been improved a lot after revision. Some comments are given below.

1. In line 467, it is said that “We obtained 30% subsamples of cells from each cluster (16,351 cells in total ).” However, in the response for referee#2-2, it is said that “The total number of cells after subsampling was 10,324. Genes with fewer than 10 cells expressed in the subsampled state were filtered out, and 16,351 genes were used as variables.” Please double check.

2. For UpSet plots (Fig. S1b&c), results are confusing. For example, in Fig. S1b, the number of interaction features of all four methods is 32, but the number of interaction features of SCAD and Lasso is 27<32. Please check.

Reviewer #3: The authors have addressed my previous questions.

**Have the authors made all data and (if applicable) computational code underlying the findings in their manuscript fully available?**

Reviewer #1: None

Reviewer #2: Yes

Reviewer #3: Yes

PLOS authors have the option to publish the peer review history of their article (what does this mean?). If published, this will include your full peer review and any attached files.

Reviewer #1: No

Reviewer #2: No

Reviewer #3: No

Figure Files:

Data Requirements:

Reproducibility:

References:

---

## [Decision Letter · Decision Letter 2]

19 Oct 2021

Dear Prof. Ohkawa,

We are pleased to inform you that your manuscript 'Discriminative feature of cells characterizes cell populations of interest by a small subset of genes' has been provisionally accepted for publication in PLOS Computational Biology.

Best regards,

Lilia M. Iakoucheva, Ph.D.

Associate Editor

PLOS Computational Biology

Ilya Ioshikhes

Deputy Editor

PLOS Computational Biology

Reviewer's Responses to Questions

**Comments to the Authors:**

Reviewer #2: The authors have addressed my previous comments.

**Have the authors made all data and (if applicable) computational code underlying the findings in their manuscript fully available?**

Reviewer #2: Yes

PLOS authors have the option to publish the peer review history of their article (what does this mean?). If published, this will include your full peer review and any attached files.

Reviewer #2: No

---

## [Editor Report · Acceptance letter]

1 Nov 2021

PCOMPBIOL-D-21-00960R2 

Discriminative feature of cells characterizes cell populations of interest by a small subset of genes

Dear Dr Ohkawa,

I am pleased to inform you that your manuscript has been formally accepted for publication in PLOS Computational Biology. Your manuscript is now with our production department and you will be notified of the publication date in due course.

With kind regards,

Zsofia Freund
